# Pharmacological Functions, Synthesis, and Delivery Progress for Collagen as Biodrug and Biomaterial

**DOI:** 10.3390/pharmaceutics15051443

**Published:** 2023-05-09

**Authors:** Nan Zhou, Yu-Da Liu, Yue Zhang, Ting-Wei Gu, Li-Hua Peng

**Affiliations:** 1College of Pharmaceutical Sciences, Zhejiang University, Hangzhou 310058, China; 3160105379@zju.edu.cn (N.Z.);; 2State Key Laboratory of Quality Research in Chinese Medicine, Macau University of Science and Technology, Macau 999078, China

**Keywords:** collagen, pharmacological functions, biosynthesis, delivery technologies

## Abstract

Collagen has been widely applied as a functional biomaterial in regulating tissue regeneration and drug delivery by participating in cell proliferation, differentiation, migration, intercellular signal transmission, tissue formation, and blood coagulation. However, traditional extraction of collagen from animals potentially induces immunogenicity and requires complicated material treatment and purification steps. Although semi-synthesis strategies such as utilizing recombinant *E. coli* or *yeast* expression systems have been explored as alternative methods, the influence of unwanted by-products, foreign substances, and immature synthetic processes have limited its industrial production and clinical applications. Meanwhile, macromolecule collagen products encounter a bottleneck in delivery and absorption by conventional oral and injection vehicles, which promotes the studies of transdermal and topical delivery strategies and implant methods. This review illustrates the physiological and therapeutic effects, synthesis strategies, and delivery technologies of collagen to provide a reference and outlook for the research and development of collagen as a biodrug and biomaterial.

## 1. Introduction

Collagen is an essential natural protein of the connective tissue that is widely found in the skin, bone, cartilage, teeth, tendons, ligaments, and blood vessels of animals. At the cellular level, collagen, as a major component of the extracellular machinery, is involved in supporting organs and protecting the body, but also participates in many cellular activities of migration, differentiation, and proliferation. Since the time of ancient Egypt, collagen was extracted from animal skin and used as glue in the therapies of medicine and health, including for burns, trauma, canthus diseases, beauty, orthopedics, hard tissue repair, wound bleeding, and so on [1]. Collagen has a promising prospect in clinical application. Since 1690, due to its excellent therapeutic efficacy in diseases, the Netherlands, UK, France, and Germany have produced collagen as a major industry. At present, recombinant human collagen III from *Pichia pastoris* has been used for anti-aging, and aortic valves made of pure collagen are already in clinical trials [2]. However, despite the in-depth study and wide use of collagen, the clinical application of collagen is seriously hindered by the limited traditional extraction strategy from mammals with low extraction efficiency, purity, and allergen problems, and more efficient production and drug delivery methods have continued to be significant issues [3]. Currently, novel total synthesis methods are being explored, but they have not achieved large-scale production due to their immature technology [2]. Alternatively, the genetic engineering of recombinant collagen has become a production method in the new era, and has the advantages of simple composition, high safety, and controllable production process, and can also avoid animal-derived pathogens [4]. Although these novel synthesis methods exhibited unique advantages, the clinical safety and low absorption efficiency of collagen products’ delivery as biomacromolecules in oral, transdermal, and injectable administration remain unresolved [5]. In this review, the frontier progress of the physiology and therapeutic functions of the different collagen types are summarized, and the synthesis and delivery strategies are highlighted to provide an outlook for developing collagen into a biodrug and biomaterial for treating various diseases.

## 2. Structural Features and Classification

Collagen is a stable fibrous protein formed by three polypeptide chains which take the same axis as the center. Owing to the interaction of amino acid residues, three polypeptide chains are closely bound together by parallel and interchain hydrogen bonds through the right-handed helix. Each α molecule of the peptide chains is composed of Gly-X-Y peptide segments (X and Y represent any amino acid residues other than Gly, usually referring to Pro and Hyp, respectively), which form a left-handed helix [3]. At present, more than 20 types of collagen have been discovered and divided into I, II, III, and other types, according to the combination of the three amino acid chains. Due to their different primary structures, they form various morphologies and show some specific biological properties and functions. The most common types of collagen in humans are types I–V [6]. Here, the subunits and their composition, tissue distribution, and other main characteristics of types I–V collagens are summarized in Table 1.

It can be seen that I, II, III, and V are the main collagen types that make up the essential tissues in bone, cartilage, tendon, skin, and muscle. The α1/α2 subunits in collagen are the basis of tissue support functions, and the α3 subunit may be the source of cellular control and site-binding functions. The three-strand peptide chain compositions of these collagens are summarized in Figure 1.

Many researchers validated that collagen I is the most important molecular species of collagen in animals. It consists of two α1(I) chains which are similar in amino acid sequences, and one α2(I) chain. The α3(I) chain plays an important role in transmission and is mostly found in the skin of some teleost fishes [7,8].

Collagen II is synthesized by animal cells. It mainly exists in human and animal skin, bone, cartilage, tendon, and other connective tissues and is also an important component of the extracellular matrix. It consists of three α1(II) chains, which is a typical triple helix structure. Collagen II forms small-diameter fibers in a cartilage matrix. These small fibers form a finely spaced network structure in cartilage tissue, which makes collagen distribute in proteoglycan to the maximum extent. This structure can provide mechanical stability for cartilage loading and is one of the main extracellular matrices in articular cartilage [9,10].

Collagen III is the main collagen in human skin, fascia, and tendons, and the ratio to collagen I is four to one. Collagen III is relatively small and exists between the epidermis and dermis and is the main component of the microcollagen layer. It is the key to propping up the epidermis. Collagen IV is an indispensable component of the cell basement membrane in the human body. Its structure is a heterotrimer spiral chain, which is composed of 3α-peptide chains. Finally, collagen V is distributed around cells and may play a bridge role between the basement membrane and connective tissue [11,12].

## 3. Physiological and Therapeutic Effects of Collagens

The main physiological function of collagen is to provide support in connective tissue as a structural protein, especially for collagen I, which has the highest content in the human body. Collagen also has many other biological functions such as binding to integrins and other factors to mediate signal transduction, to connect cells and extracellular matrix biological signals. There is a relationship between a lack of all kinds of collagen and bone disease, skin disease, gastrointestinal disease, vascular disease, kidney disease, neurological diseases, and even cancer, so the potential of collagen in clinical treatment may be still not be fully developed (Figure 2, Table 2).

### 3.1. Collagen for Treating Skin Injuries

Collagen also plays an important role in regulating the regeneration of tissues, promoting cell adhesion, and cell proliferation. Previous studies show that collagen is closely involved in cell proliferation, differentiation and migration, intercellular signal transmission, tissue formation, blood coagulation, and so on, thus it may participate in different stages of tissue repair. During hemostasis, collagen induces platelet activation and aggregation, thereby depositing fiber clots at the injured site. In the inflammatory stage, the activation of immune cells promotes the secretion of inflammatory cytokines, thus affecting the migration of fibroblasts, epithelial cells, and endothelial cells, and fibroblasts contribute to the deposition of collagen. Collagen degradation releases fragments that promote fibroblast proliferation and growth factor synthesis, leading to angiogenesis and reepithelialization. In the mature remodeling phase, the extracellular matrix is remodeled to restore its tensile strength. So, medical materials based on natural collagen are extensively used as an alternative option to traditional materials, which show non-immunogenicity, low sensitization, and better biocompatibility, and have promising clinical applications [13].

**Table 2 pharmaceutics-15-01443-t002:** Therapeutic application of collagen.

Diseases		Therapeutic Effects	Refs.
Treating skin injuries	Wound healing	Wound closure; anti-bacterial activity	[14,15,16,17,18,19]
	Burn healing	Accelerated healing and skin appendage generation	[20,21,22,23]
	Chronic wound	Faster wound healing	[24,25,26]
Treating orthopedic diseases	Osteoporosis	Improved bone mineral density; increased bone hydroxyproline content; enhanced alkaline phosphatase level	[27]
	Bone defect healing	New bone tissue forming; guided bone regeneration	[28,29,30,31]
Treating ophthalmic diseases	Corneal defects	Filled corneal defects; restored corneal curvature	[29,32]
	Keratoconus	Increased corneal rigidity; decreased interfibrillar Bragg spacing	[33,34,35,36]
Promoting nerve regeneration	Central nerve injury	Tuning NSCs; improved motor performance; reduced formation of fluid-filled cysts; impeded collapse of musculature and connective tissue	[37,38,39,40,41,42,43]
	Peripheral nerve injury	Well-organized fibers; unimpaired myelin sheath	[44]
Anti-aging	Skin anti-aging	Reduced trans-epidermal water loss and skin pore number; increased elasticity; enhanced dermal thickness and acoustic density	[45,46,47,48]

The wet strength of a collagen sponge allows for suturing of soft tissue and provides a template for the growth of new tissue. Collagen hydrogel was shown to be a potential wet wound dressing that could significantly accelerate the production of new skin appendages [49]. Collagen dressings are usually formulated from bovine, avian, or porcine collagen and are easy to apply and remove [21]. In addition, collagen dressings can be derived from marine sources. Nile tilapia is one of the most commonly cultured fish in China, and the collagen hydrogel from tilapia skin can be used as a wound dressing for the treatment of deep second-degree burns [50]. Oral administration of collagen can also be an effective treatment for wound healing. Studies related to oral jellyfish collagen peptide and salmon skin collagen peptide have shown positive effects on wound healing [20].

### 3.2. Collagen for Treating Orthopedic Diseases

Collagen also constitutes tendons connecting bone with muscle along with other tissue such as the knee and joints, which makes muscles and bones soft and elastic during exercise. Human bones are mainly made up of calcium and collagen, of which calcium salts account for 2/3 and collagen for 1/3 [49]. Osteoporosis is caused by the loss of calcium, and collagen is essential in binding calcium to bone cells and improving osteoporosis; analysis results show that collagen forms a reticular structure in bone, which increases bone stiffness and toughness. Collagen I and collagen II are the main components of hyaline articular cartilage, fibrocartilage, and fibrous tissue [51,52]. So, collagen II cannot be used only to determine whether the cartilage is hyaline, as it can also be found in fibrocartilage [53], and loss of collagen is the cause of heterogeneous genetic disease involving bone fragility deformities, blue sclera, structural shortness, hypodentinogenesis, and hearing loss [54,55]. In a study, 13-month-old mice were used to evaluate the effects of collagen materials from silver carp skin on osteoporosis, and the results indicated that collagen materials from fish could be applied to alleviate osteoporosis or treat bone loss [27]. In application, a thermal-response hydrogel composite composed of triblock PEG-PCL-PEG copolymer, collagen, and nano-hydroxyapatite was developed by ShaoZhi Fu et al. [28] and it has shown great potential in bone tissue engineering.

### 3.3. Collagen for Treating Ophthalmic Diseases

The corneal stroma is composed mainly of collagen I fibrils organized into ~300 orthogonally arranged lamellas [56]. In addition to the stroma, the corneal endothelium consists of a single layer of interconnected hexagonal cells located on Descemet’s membrane. This layer is critical for maintaining relative stroma shedding/dehydration, which is essential for corneal transparency [56]. The Descemet’s basement membrane is mainly composed of collagen IV as well as laminin, perlecan (a heparan sulfate proteoglycan), nidogen, and, to a lesser extent, collagen VIII [57,58]. Collagen is also an important component of the lens, and the lens capsule is structurally similar to Descemet’s membrane in the cornea, as it consists of a network of collagen IV and laminin bound together by nidogen and perlecan [59]. The lens capsule serves as a supporting matrix for the epithelial cells of the anterior lens and the fibrocytes of the posterior lens. Since this structure encapsulates all lens cells, it also protects them from infection. In the retina, collagen IV is an abundant component of all basement membranes [60], and collagen IV is present in a sandwich form on both sides of Bruch’s membrane, with the middle layer containing elastic fiber transactions that can also be detected in the extracellular matrix surrounding human RPE cells [61].

Continuous supplementation of collagen peptide may improve the phenomenon of blurred vision and effectively prevent cataracts, iritis, and other ophthalmic diseases [5]. Wollensak et al. [34] validated that the crosslinked corneal collagen obtained by riboflavin and UVA irradiation increased the hardness and strength of corneal stroma and prevented the further development of keratoconus. Alternatively, some contact lenses made of collagen have been developed, which can improve the water-holding capacity of the eyes.

### 3.4. Collagen for Promoting Nerve Regeneration

Some results of molecular and cellular experiments show that collagen IV in the peripheral extracellular matrix is the key signaling molecule to activate the axon clusters. Collagen VI fulfills a neuroprotective function in the extracellular milieu of the brain that counteracts Aβ-induced neurotoxicity. This function is dependent on the assembly state of collagen VI and may involve interactions with Aβ as well as molecules on the neuronal surface. Collagen VI-related mechanisms may represent novel targets for protective strategies against AD and are supported by NIA and NINDS [60]. Collagen IV can interact with the FNIII domain of neural cell adhesion molecule 1 to improve the stability of the latter in the axial membrane and inhibit the lysosomal degradation pathway of neural cell adhesion molecule 1, to promote axonal assembly. The in vivo rat model of sciatic nerve defects showed that increasing the content of collagen IV in the regeneration microenvironment could promote the regeneration of parallel and orderly nerve bundles. At the same time, the projection accuracy and functional recovery of the regenerated nerve were improved [62]. In vitro studies by Peiwen Chen et al. [63] identified collagen VI as a novel regulator for peripheral nerve regeneration by modulating macrophage function. In practice, a study by Dan Lv et al. [64] confirmed that polycaprolactone/collagen VI conduits with sustained release of collagen VI in the local microenvironment may, through triggering macrophage M2 polarization, enhance nerve regeneration. Furthermore, according to Papon Muangsanit et al. [65], rat tail collagen I gels were used as a matrix to load human umbilical vein endothelial cells, and the tissue-engineered constructs containing aligned endothelial cells within the collagen matrix could be good candidates to treat peripheral nerve injury.

### 3.5. Collagen for Anti-Aging

Collagen in the dermis makes up 70% of the skin composition, it achieves the adhesion of myofibrin in myosin, and can also reach the basal layer of skin through blood circulation to strengthen the skin protection function, elasticity, and firmness; its proper elasticity and hardness facilitate movement of the body [66]. Moreover, the basement membrane of the epidermis is tightly combined with the collagen in the dermis, wrapping the sebaceous glands and hair root cells in dermis, thereby propping up the epidermis and dermis and reducing skin wrinkles.

Due to high biocompatibility with the human body, collagen I is the most used collagen in cosmetic production. In recent years, oral collagen supplements have become very popular. According to clinical trials in recent years, oral supplementation of collagen has improved skin hydration, roughness, elasticity, and density, and significantly reduced fragmentation of the collagen network [67,68]. A study by Chen et al. [69] showed that after continuous intake of collagen peptide for 12 weeks, the treatment group had significantly reduced periorbital wrinkles, increased facial skin moisture, and enhanced skin elasticity.

## 4. Strategies for Synthetic Collagen Production

Currently, the extraction methods include enzymatic, acid, alkali, salt extraction, and the hot water extraction method. Among them, in alkali and salt extraction methods, the collagen fibers of raw materials are dissolved in an acidic, alkaline, or neutral salt solution to destroy the intermolecular bonding force, however, these solutions are non-ideal in dissolving the tightly crosslinked collagen, thus leading to a low extraction rate [2,3]. Alternatively, enzymatic extraction is a common method of degrading collagen under certain external environmental conditions by different proteases, cutting off terminal peptides, and finally extracting collagen. The commonly used enzymes include alkaline protease, neutral protease, pepsin, papain, and so on. The method has the advantages of the fast hydrolysis reaction, no environmental pollution, high purity, and stable physical and chemical properties. It should be noted that components other than collagen such as antigenic components and lipid foreign proteins must be removed to avoid allergic reactions in the application, however, none of the strategies above can avoid the problem. Although, to avoid human or zoonotic pathogens, extracting collagen from marine organisms has been gradually adopted, unfortunately, due to the lack of hydroxyproline, collagen from marine organisms has low thermal stability, which limits its scale of production and applications [70]. Strategies of extracting collagen from natural materials urgently need to be optimized or replaced.

### 4.1. Total Synthesis Strategy

Although the semi-synthetic strategy is relatively simple, due to the limitation of material sources and strict production conditions, a more suitable method for large-scale production is urgently needed. Trimeric structures of synthetic Gly-X-Y repeats, referred to as collagen-mimicking sequences, collagen-like peptides, or collagen-related peptides, are at the forefront of scientific research to address issues associated with animal-extracted collagens, cell-produced collagen, and recombinantly synthesized collagens. The design of collagen mimetic peptides is based on the sequence of natural collagens, where the X position is predominantly occupied by proline and the Y position is most commonly 4R-hydroxyproline, a post-translationally modified amino acid with a hydroxyl group on the g-carbon of the proline side chain (single letter code O and the three letter code Hyp) [6,71]. On account of 4R-hydroxyproline not being encoded in the standard genome and the systems available for protein biosynthesis lacking prolyl hydroxylase, the enzyme responsible for the transformation, collagen mimetic peptides have been mostly produced by chemical synthesis. Currently, collagen mimetic peptides are usually synthesized using standard *N*-(9-fluorenyl) methoxycarbonyl-(Fmoc)-based SPPS, including the use of benzotriazole coupling reagents and piperidine for Fmoc deprotection. To maximize the yield of difficult couplings, particularly the sequential coupling of amino acids, some modifications are required. This can be addressed by coupling triplets instead of amino acids, increasing the amino acid coupling time, or utilizing a mixture of diaza(1,3)-bicyclo undecane and piperidine during the Fmoc deprotection steps. However, synthetic collagen mimetic peptides in dry environments are mostly amorphous aggregates, and the ideal fibers fail to be replicated by using hydrated techniques [6]. There are few reports on heterotrimeric systems that assemble beyond the triple helix, which requires furthering of our understanding of the structure and stability of heterotrimeric collagens, thus providing a blueprint for a new generation of biomimetic biomaterials.

### 4.2. Strategies for Recombinant Collagen Production

Compared with animal-derived collagen, the genetic engineering of recombinant collagen has become a production method in the new era, and has the advantages of simple composition, high safety, and controllable production process (Table 3, Figure 3). The recombinant collagen is mainly obtained by transferring the constructed DNA sequence to different hosts. For the construction of the system, the target gene is connected to the vector and then transferred to different host cells, and then recombinant collagen is induced by fermentation [72]. Numerous primary and immortalized cells have been used as host cells over the years for the production of various collagen types (primarily collagen I from fibroblasts and collagen II from chondrocytes) from various species. Based on literature research, these recombinant collagens have been produced in mammalian cells, insect cell cultures, *yeast*, and mostly in plant cell culture, and some plant-derived recombinant collagens have been reported using tobacco, transgenic maize seed, and barley. In addition, many recombinant expression systems, such as *E. coli*, *yeast*, animal cells, and plant cells, have been applied in producing essential L-ascorbic acid for the synthesis of collagen, which cannot be synthesized in humans, guinea pigs, primates, and other species.

#### 4.2.1. Protein Engineering in Animals

Recombinant collagen can also be expressed in animals, as John D C et al. [73] and Toman et al. [74] reported that mice were transformed with collagen gene fragments and αS1 casein-specific promoter gene fragments, and human collagen I was produced from the mammary glands of transgenic mice. However, amino acid analysis found only 50% of the normal level of hydroxyproline and the method also has the drawbacks of animal-derived collagen mentioned above. Tomia et al. [75] constructed a vector and adopted the method of gene implantation to secrete and express human collagen III fragments through the silk glands of a transgenic silkworm. Later, Adachi et al. [76] improved it and solved the problem of inadequate hydroxylation of proline, but the fragment still could not form a triple helix structure. It should be noted that the animal system is expensive and difficult to commercialize, while the microbial fermentation system has low cost, short cycle, easier cultivation, and easier commercial production. Therefore, research on synthesizing recombinant collagen is mainly focused on microbial systems.

#### 4.2.2. Protein Engineering in *Escherichia coli*

At present, the *Escherichia coli* (*E. coli*) expression system contains many host bacteria and carriers, which can adapt to different expression processes. It can be cultivated at a large scale and has strong anti-pollution ability. The main process of the method is introducing designed collagen target genes and vectors into *E. coli* to induce expression of the target protein. The common vectors include pGEX and pET series. Rutschmann et al. [77] expressed hydroxylated recombinant human collagen III in *E. coli*. The recombinant protein has good biocompatibility and can promote the growth of umbilical endothelial cells. Zhang et al. [78] expressed recombinant bacterial collagen, which was highly expressed in the *E. coli* system (0.1–0.12 g/L) and had no immunogenicity and no cytotoxic effect, which could effectively promote bone defects. Compared with natural collagen, the recombinant human collagen obtained by *E. coli* fermentation has the characteristics of a single component, controllable process, short production cycle, controllable product, and no potential virus. Therefore, *E. coli* expression accounts for 40% of more than 400 recombinant protein expression systems in clinical use. It has been applied to medical fields such as artificial blood vessels, hemostatic materials, and skin wound repair.

**Table 3 pharmaceutics-15-01443-t003:** Production strategies and yields of different expression systems for collagen.

Expression System	Host for Transfection	Synthetic Conditions	Productivity	Limitations	Ref.
Transgenic plants	Tobacco, corn, barley, etc.	Transforming collagen gene into plants such as transgenic tobacco and corn for expression	20 g/L	Its expression quantity and purification steps need to be improved	[79,80,81,82,83]
*Escherichia coli*	*E. coli*	Designed collagen target genes and vectors in *E. coli* to induce expression of the target protein	0.1–0.2 g/L	Affected by pH, temperature, dissolved oxygen, acetic acid concentration, carbon source, nitrogen source, etc.	[77,84]
*Yeast*	*Pichia pastoris*	Eukaryotic expression system can ensure the post-translational modification of collagen, including glycosylation, etc.	0.7–1.5 g/L	Affected by pH, temperature, and methanol content	[85,86,87]
Transgenic animals	Glands of mice and silkworms	Construction of the vector and animals are transformed with collagen gene fragments	8–20 g/L	Low level of hydroxyproline and incomplete triple helix structure	[73,74,75,76]

However, due to the prokaryotic expression of *E. coli*, the target protein often forms insoluble inclusion bodies in the cell during the expression of eukaryotic protein, which has no biological activity. In addition, the post-translational processing modification system is not perfect, the pyrogen produced makes it difficult for collagen to be used in the clinic, and the biological activity of the expressed product is not ideal. There is no verification for specific activity in the literature about the synthesis of recombinant collagen by *E. coli*, thus requiring further experimental exploration.

#### 4.2.3. Protein Engineering in *Yeast* Expression System

Compared with the *E. coli* expression system, the eukaryotic *yeast* expression system achieved many breakthrough results and ensures sufficient post-translational modification of collagen by engineered bacteria, including glycosylation, hydroxylation, etc. [85], and results in better properties. Prolyl 4-hydroxylase (P4H) is believed to play a central role in the biosynthesis of all collagen, as 4-hydroxyproline residues are essential for the folding of the newly synthesized polypeptide chains into a triple helical molecule [88]. A large number of studies have shown that the co-expression of collagen and the key enzyme prolyl 4-hydroxylase (P4H) in host cells is a better strategy. The co-expression strategy not only promotes the correct and high-level assembly of P4H subunits expressed by the host into an active tetramer but also allows collagen to be fully hydroxylated by P4H [86]. Studies have shown that co-expression with collagen can also significantly increase the half-life of P4H. For example, co-expression with collagen III can significantly increase its half-life by 15 times; the collagen obtained by co-expression of P4H in *yeast* can fold into the correct spatial structure and can be fully hydroxylated. Although the expression amount reached 0.2–0.6 g/L, it can only accumulate in the endoplasmic reticulum cavity of the cells [87]. In recent years, with the completion of fermentation methods and the emergence of methanol-regulated promoters, *Pichia pastoris* has been widely used with its unique advantages and potential. *Pichia pastoris* is a type of methanol-trophic *yeast* that can use methanol as the sole source of carbon and energy. In general, although the *yeast* preparation system can obtain high productivity, high quality, and low cost in the synthesis of collagen, it also has some limitations. Studies found that recombinant *Pichia pastoris* could not secrete and express human-like collagen in part of the medium, and complex components such as peptone and *yeast* powder should be added, which requires a specific culture medium. The secreted human-like collagen is easily hydrolyzed in the fermentation broth. It is greatly affected by the methanol concentration and the environment should be strictly controlled. For these problems of various parameters, a recent trial used tolerogenic epitopes to facilitate the production of type II collagen in *Pichia pastoris* [85], where pro-collagen monomers accumulated inside *P. pastoris* cells, whereas tolerogenic epitopes were assembled into homotrimers with stable conformation and secreted into the supernatants. In the culture and induction of *P. pastoris* strains, expression was induced in a buffered minimal methanol medium containing 1% casamino acids, and methanol was added every 12 h to a final concentration of 0.5%. Fortunately, we can set conditions according to the methods described in the Manual Version of the Pichia Expression Kit. Moreover, regarding optimal conditions of synthesis strategies in *Pichia pastoris*, to prove that recombinant pC- and pN-collagen as well as CTE could assembly into a stable triple helix, the thermal stability of the pepsin-resistant recombinant CCII was studied by digestion with a mixture of trypsin and chymotrypsin at various temperatures and an optimal temperature of 38~40 °C was obtained. However, the fundamental question underlying synthesis of collagen consists of which genes are essential or non-essential for subunit expression as well as for cellular assembly, and studies could revisit the issue of whether pro-peptide and telopeptide domains play crucial roles in the molecular assembly of triple helices in hydroxylated rCCII.

#### 4.2.4. Protein Engineering in Plants

Plants are already being used to produce antibodies, vaccines, growth factors, and many other proteins of pharmaceutical importance. Collagens that are expressed from plant cells such as transgenic tobacco and corn can avoid animal pathogens and easily bind foreign genes [89]. Wang et al. [79] constructed an expression system in corn using CGB (a recombinant full-length human collagen I, rCIa1) and CGD (carrying rCIa1 gene and a and b recombinant human P4H, rP4Ha, and rP4Hb), and expressed hydroxylated recombinant human collagen Iα1 chains. Then, callus and T1 seeds were screened by enzyme-linked immunosorbent assay for initial transgene expression detection. The results showed that the embryo-specific expression of rCIa1 using the maize globulin-1 gene promoter resulted in an average yield of 12 mg/kg of full-length rCIa1 in seeds without co-expression of rP4H and a yield of rP4H in seeds co-expressed with rP4H of 4 mg/kg rCIa1 (rCIa1-OH).

According to Zhang et al. [80], a synthetic vector was introduced into corn immature embryos through the Agrobacterium transformation method, then regenerated to obtain transgenic plants, and expressed recombinant human collagen I α1chains. The starting material is grain, where the expression of collagen I is directed by an embryo-specific promoter. Purification involves extraction at low pH, followed by the membrane and chromatographic steps to separate collagen I for characterization. The amino acid composition and immunoreactivity of collagen I are similar to those of natural human collagen I and collagen I produced by the *yeast Pichia pastoris*.

Eskelin et al. [82] expressed collagen with a designed composition and structure through recombinant DNA technology. Barley seeds were selected as the production host for recombinant full-length collagen Iα1 and related 45 kDa collagen I fragments. The expression of the 45 kDa collagen I fragment targeting the endoplasmic reticulum was controlled by three different promoters (constitutive maize ubiquitin, seed endosperm-specific rice gluten, and germination-specific barley alpha-amylase fusion) to compare their effects on collagen I cumulative influence. Results showed that the highest accumulation of 45 kDa collagen I was obtained with the gluten promoter (140 mg/kg seed), while the lowest accumulation was obtained with the α-amylase promoter. To induce homozygosity for stable production of 45 kDa collagen I in transgenic lines, double haploid progeny were produced by microspore culture. The 45 kDa collagen I expression level obtained from the best DH strain was 13 mg/kg dry seed under the ubiquitin promoter and 45 mg/kg dry seed under the gluten promoter. Mass spectrometry and amino acid composition analysis of the purified 45 kDa collagen I fragment showed that a small part of the proline was hydroxylated with no additional detectable post-translational modification. Among them, the yield of the 45 kDa fragment reached 150 g/(hm)^2^. This work demonstrated for the first time that barley seeds can be used as a collagen-related structural protein production system by using collagen I as a model and it already has commercial application prospects.

Merle et al. [83] successfully expressed hydroxylated homotrimeric recombinant collagen by co-transforming the human collagen I gene and chimeric P4H corresponding gene into tobacco plants. It was the first time that transient expression technology had been used to co-express modified enzymes derived from animal cells in tobacco, thereby improving the quality of recombinant proteins in plants. Stein et al. [81] successfully co-expressed two human genes encoding recombinant heterotrimer collagen I with human P4H and lysyl hydroxylase in tobacco plant enzyme 3 (LH3), which are responsible for key post-translational modifications of collagen. It is worth noting that plant-derived collagen I forms a heat-stable triple helix structure and exhibits similar biological functions to human-derived collagen, supporting the binding and proliferation of endothelial progenitor-like cells derived from adult peripheral blood.

The safe method of producing and purifying collagen from plants provides a broadened potential application of human recombinant collagen in regenerative medicine. Although its expression quantity and purification steps may be inferior to the *yeast* system and need to be improved, molecular farming and bioreactors for plants have great potential, offering practical, biochemical, economic, and safety advantages compared with conventional production systems [90].

Several kinds of recombinant collagen with both native and collagen-derived structures have been successfully developed and designed for production. However, to date, only a few recombinant collagen products have been used in clinical practice. The main reasons may include the difficulty of manufacturing technology and the complexity of collagen macromolecular structure, which make it difficult for enterprises to produce recombinant collagen on a large scale. At present, the collagen that can be prepared on a large scale by recombinant technology is mainly single-stranded collagen. Single-stranded collagen has a more flexible self-assembly form but is easy to degrade, while triple-stranded helix collagen fiber has better biological properties, but its synthesis has more requirements. The application of collagen in tissue engineering is limited by its low level of post-translational modification, glycosylation, and hydroxylation. It is believed that with the rapid development of synthetic biology and protein engineering technology, the exploration of efficient collagen synthesis pathways, specific modification key enzymes, and fermentation and purification process optimization are expected to promote the research and development of various types of recombinant collagen with low cost and high modification level.

## 5. Application and Delivery Strategies of Collagen for Diseases

Since collagen has low immunogenicity, good biocompatibility, and biodegradability, it is widely used as biological material in wound assembly, biological patches, bone repair materials, sutures, tissue engineering scaffolds, etc. [91]. To explore the advantages in drug delivery, collagen can also be made into novel preparations, which are combined with other auxiliary materials and drugs to find effective targets and play a therapeutic role (Figure 4).

### 5.1. Collagen Peptide and Solution

As a hydrophilic protein macromolecule, collagen can be directly administered in solution by oral or topical application. The peptide produced by collagen hydrolysis can be used as a dietary supplement for anti-aging. It should be noted that collagen is stable enough to persist or act in the body for a prolonged period of time, and it has already been confirmed that the triple helix of collagen is resistant to pepsin, thus ensuring that enough collagen reaches the intestinal tract. Subsequently, collagen can be absorbed by the intestinal tract mainly through metabolism of intestinal microbiota at a suitable weak alkaline pH. Of course, the immunogenicity of collagen products should be optimized in advance. In addition, many promising preparations have been used for the protection and oral application of collagen, such as nanoparticles acting as crosslinking agents for collagen stabilization as well as functionalized carriers for crosslinking to collagen scaffolds for novel applications [92].

Although the use of collagen peptides and the efficacy of oral collagen have been questioned, in recent years there has been some evidence supporting the ability of collagen peptides to increase skin collagen density, moisture content, and hydration within 10 days [93], while long-term oral administration of low-molecular-weight collagen peptides increased skin elasticity and reduced the average roughness of wrinkles in cheeks and the canthus [68,94]. Although there are many different types of objective measures, it is not clear how each measure translates to clinical appearance or why each is affected by collagen supplements. There is also no reliable evidence that orally digested collagen is preferentially localized to the dermis rather than to other parts of the body.

Collagen peptides have also been used in wound healing. In a rat wound model [20,95] and rabbit scald wound model [96], collagen peptide accelerated the wound closure and improved tissue regeneration at the wound site, improved angiogenesis, and increased organized collagen fiber deposition.

In addition, it has also been reported that collagen peptides can promote cartilage tissue regeneration to a certain extent, for example, in horse adipose-derived stromal cells, increased glycosaminoglycan expression and induced chondrogenic differentiation were found [97] and chondroprotective effects were found in rabbit osteoarthritis model [98].

Additionally, injectable collagen solution is a typical collagen product. It is obtained by treating collagen with various physical and chemical methods. Utilizing the high hydrophilicity of injectable collagen solution, water-soluble drugs such as hormones and insulin are gradually accommodated in the gap of collagen fibers after contact with them, so that collagen becomes a “drug reserver” and can have a sustained release effect [99,100]. Antigenicity in injected collagen calls for special attention. It has been proved at the molecular level that the main antigenic sites of injectable collagen are located in the carbon terminal and nitrogen terminal regions of the molecule, which should be selectively hydrolyzed or removed in the process of collagen hydrolysis for preparing injectable collagen solution.

### 5.2. Collagen-Based Drug Delivery Systems

The natural healing process can be complicated due to a large area, damage of vasculature and bacterial infections in the full thickness skin defects of diabetic ulcers, third-degree burns, physical trauma, or surgical procedures (including tumor resections) [101]. Although skin grafting can be used to protect wounds and promote healing, the materials are in short supply [102]. As such, many collagen hydrogels have been developed to address the requirement for tissue regeneration owing to their natural sources in the body as well as their bioactive properties. Recently, researchers created advanced collagen-based composite materials that can be easily handled and avoid significant contraction to improve the poor mechanical properties of collagen-only hydrogels and maintain coverage of the injury site throughout the healing process.

Collagen scaffolds such as collagen sponges are suitable for short-term delivery of antibiotics in the wound bed. Sponges soaked with solutions of gentamicin, cefotaxim, fusidic acid, clindamycin, or vancomycin release 99.9% of these antibiotic agents after 2 days in vitro [103]. Local infection was contained by a gentamicin-containing collagen matrix placed on a septic focus in rat abdomens, which reduced local infection for at least 3 days [103]. These sponges do not exhibit any unwanted side effects and are absorbed into tissue after a few days [104]. On the basis of a simple collagen scaffold, drug–polymer–collagen blends are promising in the improvement of characteristics. Hall Barrientos IJ et al. developed electrospun collagen-based nanofibers and examined the effects of collagen type I in them for tissue engineering applications [105]. The results showed that the addition of collagen caused a decrease in average fiber diameter by nearly half and produced nanofibers. In addition, the amorphous regions within the samples, sustained release performance, and hydrophobicity were all increased, and the antibacterial studies showed a high efficacy of resistance against the growth of both *E. coli* and *S. aureus* [105].

The scope for using injectable collagen formulations for the delivery of growth factors and consequent cellular regeneration and tissue repair is vast. In a porcine model, intestinal wound repair was expedited by treatment with collagen suspensions carrying fibroblast growth factor or transforming growth factor-β (TGF-β), and wounds treated with collagen and TGF-β had increased mean (s.e.m.) breaking loads (ileal 285 (32) g·cm^−1^ versus 180 (11) g·cm^−1^, colonic 365 (59) g·cm^−1^ versus 260 (40) g·cm^−1^) in comparison with controls and the steroid-induced impairment of breaking load in intestinal wound models was partially reversed [106]. Investigations of cellular function, migration, proliferation, and differentiation in collagen gels have led to a further understanding of the mechanism and kinetics of transport, as well as the influence of growth factors, laminin, and fibronectin [107].

Collagen hydrogels present a large, uniform surface area and can serve as a drug delivery system. A common practice has been to combine natural and synthetic polymers with synergistic properties, thereby imparting higher mechanical strength to the natural polymers and biocompatibility to their synthetic counterparts. Synthetic polymers such as poly(vinyl alcohol) and poly(acrylic acid) are blended with natural polymers such as collagen and hyaluronic acid and formulated into hydrogels, films, and sponges that are then loaded with growth hormone [108]. These formulations provide a controlled release of growth hormone from the collagen hydrogel. Gels have also been formulated with atelocollagen, produced by the removal of telopeptide ends using pepsin, and used for the delivery of chondrocytes to repair cartilage defects [109].

Collagen films have been used in wound healing and tissue engineering, primarily as a barrier. Films of ~0.1–0.5 mm thickness can be cast from collagen solutions and air-dried, including ophthalmological shields. As an added advantage, films made from biodegradable materials such as telopeptide-free reconstituted collagen demonstrate a slow release of encapsulated drugs, for example, final collagen films with a size of 150 cm^2^, with loaded pilocarpine, had a release time twice as long as that of normal gel [110]. The loaded films afford easy sterilization and become pliable after hydration, without compromising their mechanical strength.

### 5.3. Collagen-Based Tissue Engineering Systems

Collagen sponges have been used to provide a scaffold for the growth of new bone to treat bone defects. The collagen sponge delivery vehicle has been used in several orthopedic applications including the filling of craniofacial bone defects and spinal fusions to treat degenerative discs [111].

Collagen scaffolds can also be applied in soft tissue healing. Recently, Xiaofeng Jin et al. [112] used an injectable collagen scaffold to deliver autologous fat cells for repairing severe vocal fold injury, and the collagen scaffold maintained the stability of implants after injection and reconstructed the vocal fold structure. More GFP-positive cells were found at day 3 and day 7 in rats with fat cells injected with collagen scaffold than the non-collagen group. Panaiy AC et al. [113] evaluated the efficacy of a novel modification of a collagen–glycosaminoglycan scaffold with autologous micrografts (CGS + MG) using a murine diabetic ulcer model, which achieved a faster healing rate, greater cellular proliferation, collagen deposition, and keratinocyte migration with higher density and greater maturity of microvessels. The wounds appeared healthy with no signs of necrosis on the wound border, exudate, or biofilm formation, and the thickness of the wound bed in CGS + MG (624.4 ± 284.3 µm) was significantly greater than in the blank mice (63.2 ± 27.3 µm), and in terms of collagen deposition, the CGS + MG group (34.5 ± 5.2%) showed significantly lower collagen deposition than the blank group (18.7 ± 2.9%). At the same time, anti-Ki-67 staining highlighted differences in the cellular proliferation. The overall number of Ki-67-positive nuclei was higher in the CGS + MG group (54.7 ± 30.0 Ki-67+/HPF) than the blank group (39.2 ± 18.9 Ki-67+/HPF).

Collagen membranes have been used for wound dressings, dural closures, reinforcement of compromised tissues, and guided tissue regeneration. Wound healing in diabetic db/db mice has been modulated by a sustained release of human growth hormone encapsulated in collagen films [114].

Collagen-based wound dressings have long been used to cover burn wounds and treat ulcers [115,116]. They have a distinctive practical and economic advantage compared to growth factor and cell-based treatment of full thickness wounds and have been formulated in several different ways (Table 1). An unconventional form consisting of powdered avian collagen is effective at expediting chronic wound healing [117]. The powder promotes cellular recruitment, activation of the inflammation phase of wound healing, and support for new tissue growth—similar in function to collagen sponges.

Cultured skin substitutes from cryo-preserved skin cells have been used to cure chronic diabetic wounds [118]. In lieu of pathological skin, the contracted collagen lattice serves as a support for epithelial cell growth and differentiation. Collagen implants have also been used in corneal healing, and corneal cells have a normal appearance when cultured individually on a synthetic collagen matrix.

Researchers have developed and used plastically compressed collagen to construct bioengineered corneal grafts [119]. The application of collagen IV as a biomaterial helps to create a natural environment and substrate for corneal endothelial cells as well as epithelial cells of the lens and retina [120].

As collagen is non-antigenic, biodegradable, hemostatic, and easily functionally modified, it is mainly used as a biomaterial. Collagen has been widely prepared into gel, sponges, tubing, spheres, membranes, rigid forms, and other biological materials for drug delivery or tissue engineering, such as wound dressing, corneal shields, bone repair materials, and skin replacement. The efficacy of oral collagen supplements for anti-aging is controversial, and a studies have tried to use it for the treatment of corneal diseases, oral ulcers, and wounds. The effect of collagen as a biological drug may be related to the drug form, molecular weight, and molecular modification, which need to be studied.

## 6. Discussion and Outlook

As a fundamental structural element of most connective tissues, collagen plays an important role in maintaining the biological and structural integrity of tissues. Its different forms and functions in different tissues have inspired biomaterial researchers to develop new biomaterials similar in structure and biological activity to collagen, especially collagen I. At present, the traditional synthetic process for collagen production is a mature industrial production method, but its cost is still high, and there are problems of immunogenicity, and the effective extraction rate of collagen cannot reach 100%. The total synthesis method can directionally synthesize the desired protein sequence by chemically synthesizing a specific trimer structure and may solve those problems above. With the continuous development of total synthesis technology, scientists have explored some simple synthesis routes, but have demonstrated difficulty in obtaining the correct folded protein structure. To solve the existing problems of the strategy, improving the hydration technology in a dry environment to synthesize the ideal fiber and conducting further basic research on the structure and stability of heterotrimeric collagens may be a potential direction.

The use of modern biotechnology to produce collagen through recombinant fermentation is easier to control and operate and avoids most exogenous substances. Among them, *E. coli* expression systems have the issue of inclusion bodies, which should be addressed [77]. The fermentation fluid of the *yeast* system may hydrolyze the product, thus strategies to inhibit protein degradation are therefore required, and it is necessary to precisely adjust the methanol content, temperature, pH, and dissolved oxygen parameters to their optimal values [85]. Therefore, solving the by-product problem and controlling exogenous substances in the synthesis process are urgently needed to optimize the process and utilize recombinant collagen. Alternatively, the plant expression system has unique potential, offering practical, biochemical, economic, and safety advantages compared with conventional production systems, but yield and expression efficiency, as well as purification technology, are crucial problems.

At the same time, collagen has proven to have a significant advantage as a biodrug or a biomaterial. As biodrugs, different drug forms such as collagen peptides, collagen molecules, and collagen-like substances are administered. Some researchers have developed specific types of nanoparticles/nanospheres/microspheres to deliver collagen. Whether oral collagen peptide has an anti-aging effect is still controversial, and more clinical evidence and mechanism research are needed. At present, transdermal or injectable strategies are mainly used to maximize the performance of collagen in drug delivery. It has been reported that collagen peptides can promote cartilage tissue regeneration, wound healing, and improvement in corneal disease.

As a biomaterial, collagen is widely used in drug delivery and tissue engineering. Collagen has been used as a carrier and developed into many novel specific types, such as collagen films, collagen shields, collagen sponges, and collagen hydrogels/gels. Transdermal transport efficiency and the appropriate administration route for collagen as a macromolecule also urgently need to be optimized. To solve the existing problems, novel preparations to convert collagen products into specific molecules that can be transported are promising. To achieve this goal, developing more methods to study the transdermal properties in depth and modifying the structure of collagen might be the priority. In addition, a combination of collagen delivery and novel physical methods may be possible, such as microneedle patches and infrared heating, etc.

It should be noted that collagen also has potential limitations in application, such as the prominent problem of the pollution of the collagen. Some cases have indicated that following the principle of proper amount and appropriateness is necessary and that excess collagen will also increase the burden on the kidneys. Moreover, the handing of collagen is difficult and needs to be improved, given that the natural crosslinking pathways of collagen do not occur in vitro. Chemical, physical, and enzymatic crosslinking methods have been used to control mechanical stability, degradation rate, and immunogenicity of the biopolymer scaffolds upon their implantation. However, to date, there is no gold standard protocol for crosslinking collagen-based materials [92]. In addition, risks in clinical applications cannot be ignored. For example, implanted type I collagen showed greater cavitations at the interface with the intact tissues as well as very dense inclusions containing large cavities which interfered with axonal growth in spinal cord-injured animals [121]. All collagen products need to undergo rigorous testing before being applied to the human body.

In general, collagen-based products have great potential for the treatment of various diseases, drug delivery, and tissue engineering. For scale application, optimizing the production process and the routes of drug delivery will be necessary. It is believed that optimizing the semi-synthesis, total synthesis, and recombinant collagen expression will provide a more efficient pathway for collagen production. The evolution of engineering and nanotechnology enable collagen materials for localized and sustained delivery of bioactive/therapeutic molecules and living cell populations, thereby expanding the clinical applications to facilitate the reuse of collagen.

## Figures and Tables

**Figure 1 pharmaceutics-15-01443-f001:**
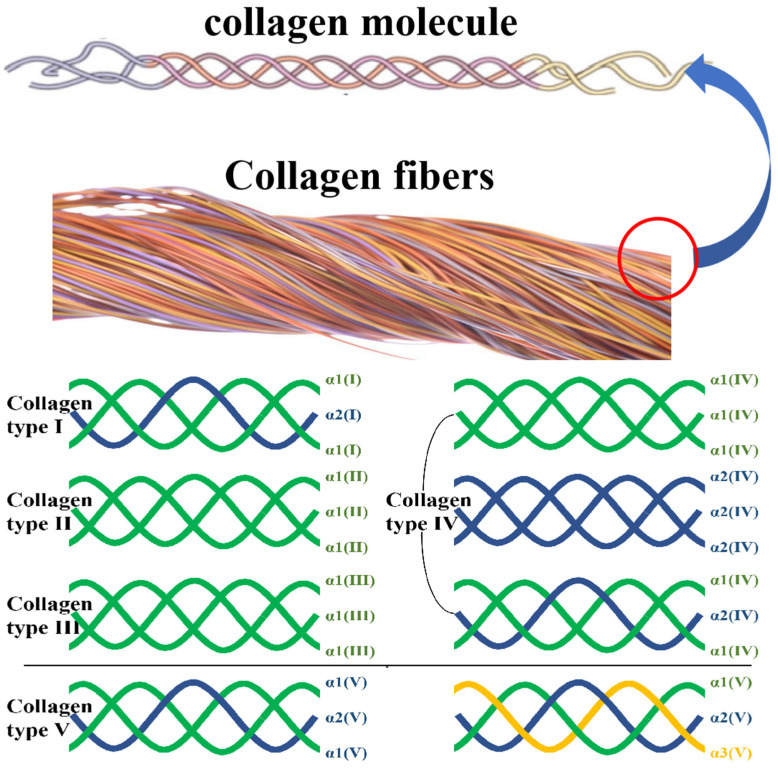
Collagen structure.

**Figure 2 pharmaceutics-15-01443-f002:**
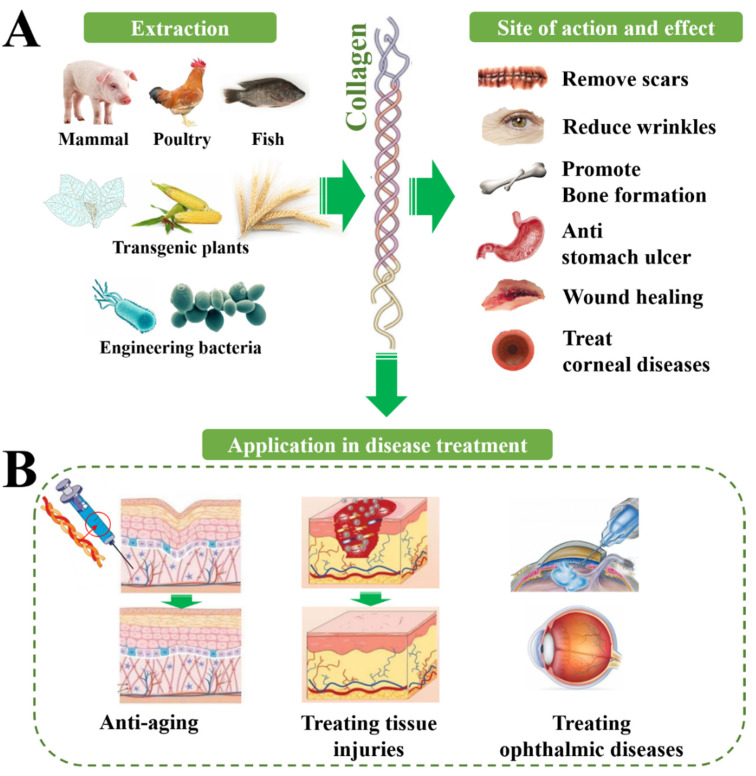
Biological functions of collagen from a variety of sources and their translation into therapeutic application. (**A**) The therapeutic effect of collagen on different sites, including promoting tissue repair, smoothing wrinkles, promoting bone regeneration, etc. (**B**) The application of collagen in the treatment of different diseases.

**Figure 3 pharmaceutics-15-01443-f003:**
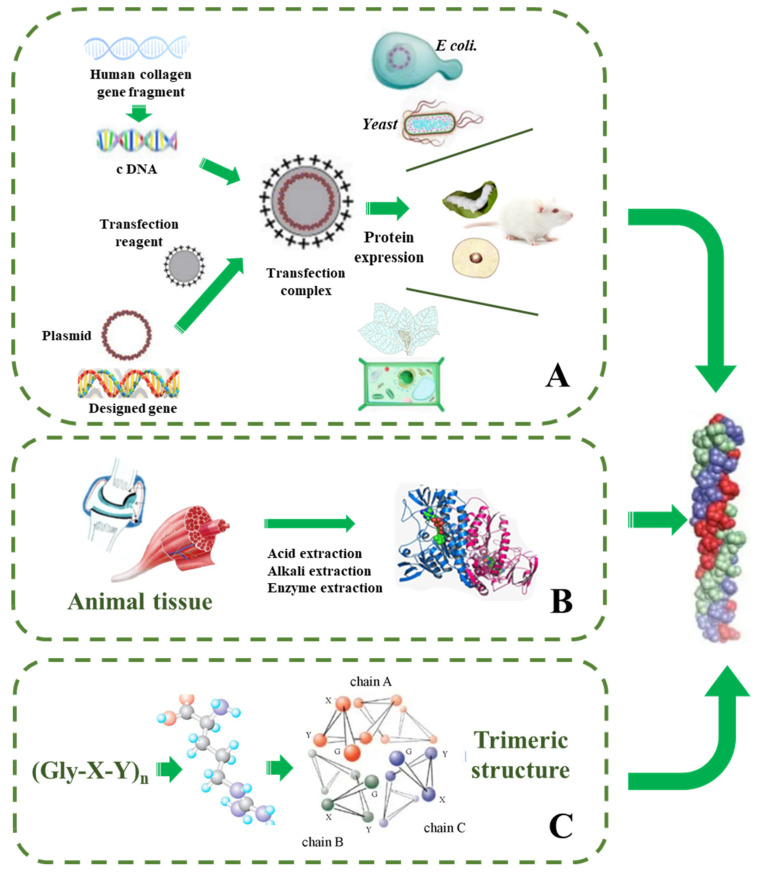
Methods of producing collagen. (**A**) Recombinant protein expression system. (**B**) Direct extraction. (**C**) Total synthesis strategy, the trimeric structure of Gly-X-Y sequence.

**Figure 4 pharmaceutics-15-01443-f004:**
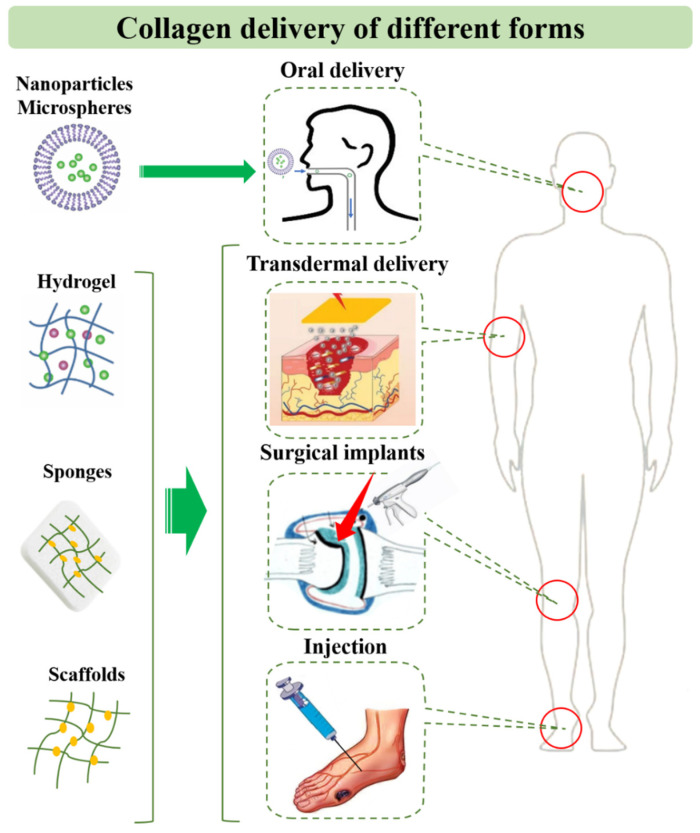
Collagen-based drug delivery strategies.

**Table 1 pharmaceutics-15-01443-t001:** Structures and characteristics of collagens I–V.

Types	Subunits and Composition	Composition of Molecular Aggregates	Tissue Distribution	Functions
Ⅰ	α1(I) × 2, α2(I)	Large-diameter cross strip fiber	Bone, cornea, skin, tendon, ligament, tumor	Support fiber
II	α1(II) × 3	Small-diameter cross strip fiber	Hyaline cartilage, vitreous body, intervertebral disc	Support fiber
III	α1(III) × 3	Small-diameter cross strip fiber	Skin, blood vessels, muscles, internal organs	Support fiber
IV	α1(IV) × 3; α2(IV) × 3; α1(IV) × 2, α2(IV)	Non-fibrous reticular structure	Basement membrane	Reticular scaffold, control of multifunctional cells, site binding
V	α1(V) × 2, α2(V); α1(V), α2(V), α3(V)	Small-diameter cross fibers, or forming molecules with type VI chains	Smooth muscle, cultured cells, embryonic tissue, peritoneum, placenta, skin, bone	Small fibers around the supporting cells

## Data Availability

Not applicable.

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
