# Peer review of "Pharmacological Functions, Synthesis, and Delivery Progress for Collagen as Biodrug and Biomaterial"

_pharmaceutics, 2023, doi:10.3390/pharmaceutics15051443_

Round 1

Reviewer 1 Report

The manuscript introduces a comprehensive overview of collagen used for various biodrug and biomaterial applications. In specific, the physiological and therapeutic function of type I~V collagen, synthesis strategies, and drug delivery were described. The authors introduce various strategies to resolve the adverse effects caused by unwanted by-products, foreign substances, and immature synthetic processes. Furthermore, the authors describe the biomedical applications of collagen scaffolds, such as collagen-based sponges, hydrogels, films, and wound dressings. This manuscript provides promising insight into developing collagen biodrug and biomaterial for treating various diseases, however, there are still some comments for this manuscript.

1) Please check all the abbreviations are defined at their first appearance in the text (e.g., CII, CI1).

2) Please check all the figures are mentioned in the manuscript (e.g., 2A, 2B, 3A, 3B, and 3C)

3) Please add references in Tables 1 and 2. And, please summarize references and include them in the manuscript.

4) The authors described the collagen scaffolds (e.g., collagen-based sponges, shields, hydrogels/gels, films, and wound dressings) for various biodrug and biomaterial applications. The reviewer thinks that these scaffolds have the potential for treating various diseases, but this manuscript did not explain how effective they are in promoting healing. Please add analysis data if there is any quantitative analysis data for each sample.

5) The authors mentioned that collagen is closely involved in cell proliferation, differentiation, migration, intercellular signal transmission, tissue formation, blood coagulation, and so on. It is required to provide additional data or explanation on the molecular relationships of collagen in more detail.

6) It is needed to explain the optimal conditions of synthesis strategies with various parameters (e.g., thermal stability, methanol content, temperature, pH, and dissolved oxygen). Also, the authors should clearly provide reasons why various parameters were used for synthetic collagen production.

7) It is recommended to add a table on the application of collagen used for various diseases.

8) The reviewer thinks that Figure 2B does not include the mechanism of collagen. Please correct the caption of Figure 2B.

9) It is required to discuss the current limitations of collagen for clinical application (e.g., short degradation time, difficult handling, lack of regenerative ability).

10) The authors mentioned that oral administration of collagen can also be an effective treatment for wound healing. The reviewer thinks that it should be noted that collagen needs to be stable enough to persist or act in the body for a prolonged period of time. Please add the explanation in the manuscript.

Reviewer 2 Report

The paper provides a comprehensive overview of collagen's physiological and therapeutic effects, synthesis strategies, and delivery technologies as a biomaterial. The diverse biomedical applications are outlined in the paper. Only a few additions are required regarding the stability and processability of collagen.

How does the pH of the intestinal tract modify the efficacy and safety of the protein? 

Regarding the nanofibrous scaffolds, the changes in the characteristics of the drug-polymer-collagen blends should be illustrated with various examples (e.g., 10.1016/j.ijpharm.2017.08.071).

After providing this additional information, the paper can be accepted for publication.
